# Skeletal Torsion Tunneling and Methyl Internal Rotation: The Coupled Large Amplitude Motions in Phenyl Acetate

**DOI:** 10.3390/molecules27092730

**Published:** 2022-04-23

**Authors:** Lynn Ferres, Luca Evangelisti, Assimo Maris, Sonia Melandri, Walther Caminati, Wolfgang Stahl, Ha Vinh Lam Nguyen

**Affiliations:** 1Institute of Physical Chemistry, RWTH Aachen University, Landoltweg 2, D-52074 Aachen, Germany; lynn.ferres@rwth-aachen.de (L.F.); wolfgang.stahl@rwth-aachen.de (W.S.); 2Dipartimento di Chimica “G. Ciamician”, Università di Bologna, Via Selmi 2, 40126 Bologna, Italy; luca.evangelisti6@unibo.it (L.E.); assimo.maris@unibo.it (A.M.); sonia.melandri@unibo.it (S.M.); 3Univ Paris Est Creteil and Université Paris Cité, CNRS, LISA, F-94010 Créteil, France; 4Institut Universitaire de France (IUF), F-75231 Paris, France

**Keywords:** internal rotation, large amplitude motions, skeletal motion tunneling, rotational spectroscopy

## Abstract

The rotational spectrum of phenyl acetate, CH_3_COOC_6_H_5_, is measured using a free jet absorption millimeter-wave spectrometer in the range from 60 to 78 GHz and two pulsed jet Fourier transform microwave spectrometers covering a total frequency range from 2 to 26.5 GHz. The features of two large amplitude motions, the methyl group internal rotation and the skeletal torsion of the CH_3_COO group with respect to the phenyl ring C_6_H_5_ (tilted at about 70°), characterize the spectrum. The vibrational ground state is split into four widely spaced sublevels, labeled as A0, E0, A1, and E1, each of them with its set of rotational transitions and with additional interstate transitions. A global fit of the line frequencies of the four sublevels leads to the determination of 51 spectroscopic parameters, including the Δ*E*_A0/A1_ and Δ*E*_E0/E1_ vibrational splittings of ~36.4 and ~33.5 GHz, respectively. The *V*_3_ barrier to methyl internal rotation (~136 cm^−1^) and the skeletal torsion *B*_2_ barrier to the orthogonality of the two planes (~68 cm^−1^) are deduced.

## 1. Introduction

Phenyl acetate (CH_3_COOC_6_H_5_, also called phenyl ethanoate, phenol acetate, acetyloxybenzene, or acetoxybenzene) is a colorless liquid with a plastic-like odor, often used as a solvent in chemical reactions. It can be synthesized by adding acetic anhydride to phenol [1], via Baeyer–Villiger oxidation of acetophenone [2], or by decarboxylation of aspirin [3]. Currently, phenyl acetate is a drug being studied in the treatment of cancer [4]. Naturally occurring in mammals, phenyl acetate induces differentiation, growth inhibition, and apoptosis in tumor cells. Despite the great interest in medicine, knowledge of its molecular structure is very limited, though it might be helpful to develop force fields to study biological processes at a molecular scale towards understanding how phenyl acetate affects tumor cells. 

Many acetates have already been studied using rotational spectroscopy, notably a series of linear aliphatic acetates starting from methyl acetate to hexyl acetate [5,6,7,8,9,10]. The methyl group in the acetyl moiety undergoes internal rotation, causing each rotational transition to split into an A-E doublet. The torsional barrier is almost invariant at approximately 100 cm^−1^ [7]. If a C=C double bond is attached at the *α*,*β*-position counting after the oxygen atom, the value increases to about 150 cm^−1^, as in the case of vinyl acetate [11,12] and butadienyl acetate [13] where the double bond(s) are located in the OCO-plane. Surprisingly, this is not the case in isopropenyl acetate, where we observed an unusual C_1_ structure. The isopropenyl group is tilted out of the OCO-plane, and a torsional barrier of 135.3498(38) cm^−1^ was deduced, a value which lies between 100 and 150 cm^−1^ [14]. Since the molecular structure of phenyl acetate is very similar to that of isopropenyl acetate, we were interested in answering two questions: (i) Does phenyl acetate have a C_s_ or C_1_ symmetry? (ii) How high is the barrier to internal rotation of the methyl group of phenyl acetate?

Though in many cases, studies on medium-sized molecules containing a phenyl ring have reported a planar heavy-atom structure as the most stable one, such as anisole [15], phenetole [16], methyl salicylate [17], or acetophenone [18], there are several molecules where at least one conformer observed in the rotational spectrum has C_1_ symmetry. For example, the CH_2_OH group in benzyl alcohol [19] as well as in its derivative 3,5-difluorobenzyl alcohol [20] is tilted out of the phenyl ring. In the case of 2-phenylethylamine, none of the observed four conformers possesses a structure where all heavy atoms are within the aromatic plane [21,22]. In *cis*-acetanilide, C_6_H_5_(NH)(CO)CH_3_, Cabezas et al. reported doublets with splittings in the order of a few tens of kHz [23]. They were interpreted to be due to the tunneling motion of the phenyl ring between two equivalent non-planar conformations, with the phenyl ring tilted out of the (NH)(CO) plane by about 40° through a transition state where the acetamide group and the phenyl ring are perpendicular. Aviles Moreno, Petitprez, and Huet observed similar splittings in the order of about 100 kHz for *E*-phenylformamide, where the phenyl ring is tilted out of the (NH)(CO) plane by the same angle [24]. These splittings arise from Coriolis interactions, from which the Δ*E*_01_ value was determined to be 3.732027(43) GHz. In phenyl formate [25], the phenyl ring tilt angle of 72° is remarkably larger than in *cis*-acetanilide and *E*-phenylformamide, and the ring also undergoes a tunneling large amplitude motion (LAM). The energy levels of molecules featuring such an inversion motion split into a (+) and a (−) state due to a symmetric and an anti-symmetric wave function, separated by Δ*E*_01_, for example in ammonia [26] (see Figure 1) [27]. In phenyl formate, the Δ*E*_01_ value of 46.2231(25) GHz is large, making the (+)←(−) and (−)←(+) transitions non-observable in the frequency range accessible by the spectrometer in use. However, the theoretically unsplit (±)←(±) transitions still show small splittings, referred to as v_t_ = 0 and v_t_ = 1, due to Coriolis interactions, which were observed and could be fitted with the *spfit* program [28], but many complications occurred in further assignments and fitting processes that relied on a small number of the less intense v_t_ = 1 transitions and Coriolis splittings between the v_t_ = 0 and 1 states.

Regarding the closely related forms of phenyl acetate and phenyl formate, we supposed a similar structure with the phenyl ring and the O-(C=O)-C plane tilted with respect to each other, enabling a skeletal tunneling motion. However, phenyl acetate features an additional methyl internal rotor, which further splits the rotational signals into an A and an E symmetry species. Thus, the A-E pairs are expected for each transition of the torsional levels, resulting in a total of four species called A0, E0 (v_t_ = 0), and A1, E1 (v_t_ = 1). Note that these sublevels could be labeled more properly as A^+^, A^−^, E^+^, and E^−^. The splittings are shown in the energy level scheme in Figure 1. To fit transition frequencies belonging to different phenyl tunneling states, the *spfit* program is well-suited. Some of its main advantages are the possibility for users to build up an individual Hamiltonian, the ability to define any desired parameters in the input, and the short calculation time. Unfortunately, *spfit* can only fit the A and E torsional transitions separately, and deriving the torsional barrier is a tedious task in low barrier cases [29]. The program *XIAM* is a counterpart, which is specialized for fitting the A-E splittings due to three-fold methyl internal rotation but is incapable of treating tunneling states arising from a two-fold potential [30]. A further separate fit was performed with the *XIAM* program where only the A0–E0 doublets are considered to alternatively access the methyl torsional barrier.

## 2. Results

### 2.1. Quantum Chemical Calculations

Using the *Gaussian* 16 program package [31], the molecular structure of phenyl acetate was optimized at the MP2/6-311G++(d,p) level of theory, revealing a non-planar structure with the methyl group *entgegen* with respect to the phenyl ring, called conformer I, illustrated in Figure 2. A second conformer, with the methyl group *zusammen* with respect to the ring, was calculated much higher in energy (1347 cm^−1^) and was not considered in the present work. Similar to the case of phenyl formate, the phenyl ring is tilted out of the O-(C=O)-C plane by about 72° for conformer I. Due to the symmetry of the phenyl ring and the planarity of the acetyl moiety, four equivalent minima are possible, as shown in Figure 3, where the dihedral angle *α* = ∠(C_13_-O_12_-C_3_-C_2_) was varied in 10° steps, while all other geometry parameters were optimized. The obtained energy points were parametrized using a Fourier expansion, including terms with the correct symmetry of *α*, which are collected in Appendix A. The two pairs I_a_/I_a_^*^ and I_b_/I_b_^*^ describe two double minima in the potential energy curve, separated by a barrier of 20.65 cm^−1^, while I_a_ and I_b_, as well as I_a_^*^ and I_b_^*^, can be converted into each other by rotating the phenyl ring by 180° with a much higher conversion barrier of 837.45 cm^−1^. The geometry of the local maximum between I_a_ and I_a_^*^ (or I_b_ and I_b_^*^) was also optimized as a transition state, yielding a slightly higher barrier of 26.71 cm^−1^. The dihedral angle *β* = ∠(C_3_-O_12_-C_13_-C_14_) corresponds to the orientation of the acetyl group and stays nearly invariant at 180° in the *entgegen* configuration, which is known to be much more stable than the *zusammen* one (*β* = 0°). The Cartesian coordinates of the I_b_ structure are given in Appendix A. The rotational constants are *A* = 3592.4315 MHz, *B* = 813.8922 MHz, and *C* = 744.3161 MHz; the three dipole moment components are *μ_a_* = −0.41 D, *μ_b_* = 0.90 D, and *μ_c_* = −1.46 D. Therefore, mainly *b*- and *c*-type transitions are expected in the rotational spectrum, eventually accompanied by some weak *a*-type signals. Frequency calculations were carried out at the same level of theory, yielding an imaginary frequency describing a ring bending motion. This observation has been frequently reported in calculations using the MP2 method [32]. Repeating the geometry optimization and frequency calculations at various levels of theory shows that most levels state no imaginary frequency.

The dihedral angle γ = ∠(O_15_-C_13_-C_14_-H_16_) was varied in a 10° grid to calculate the torsional barrier of the methyl internal rotation. The potential curve is given in Figure 4. From the Fourier parameterization (see Appendix A), we determined a *V*_3_ term of 113.18 cm^−1^. A very similar value of 113.40 cm^−1^ was deduced by optimizing the maxima as transition states [33].

To study the coupling between the two LAMs, we calculated a two-dimensional potential energy surface (2D-PES) depending on *α* and *γ*, as shown in Figure 5. The obtained energy points were also parameterized using a Fourier expansion, including terms with the correct symmetry of *α* and *γ* given in Appendix A. Along the *γ*-axis, the threefold symmetry of the methyl group can be clearly seen. For a given *γ* minimum, for example, *γ* = 0°, the four equivalent structures along the *α*-axis are also recognizable. The minima regions of this 2D-PES are very broad and flat. Therefore, more colors were used in the lower 50% area to enable the distinction of the four minima.

At this point, it is clear that the situation observed for phenyl acetate is very similar, for the skeletal torsion, to that of phenyl formate [25]. The tunneling LAM of the skeletal motion causes low-lying excited torsional states, which are most probably observable in the rotational spectrum. To calculate these tunneling states, a two-rigid-top model was applied, as described in Ref. [25], using the following Hamiltonian:(1)$$H=-F\frac{d^{2}}{d\alpha^{2}}+V_{0}+\sum_{n=1}^{7}V_{2n}\cos\left(2n\alpha \right).$$

The effective torsional constant *F* of 0.32635 cm^−1^ is calculated as *h*/8π^2^*Ic,* where *I* = 51.654 uÅ^2^ is the effective moment of inertia and *c* is the speed of light. *I* is calculated from the moments of inertia of the phenyl top *I_p_* = 121.034 uÅ^2^ and of the acetyl top *I_ac_* = 90.112 uÅ^2^. Both were taken from ab initio. The Fourier expansion terms are those obtained from the parameterization of the potential energy curve given in Figure 3. Direct diagonalization of the built-up Hamiltonian matrix yields the following eigenvalues, which represent the energies of the degenerate torsional states given in parentheses: 13.801 (0), 14.823 (1), 31.931 (2), and 42.330 (3) cm^−1^. The potential function in the range from 50° to 130°, including the torsional energy states, is illustrated in Figure 6.

The difference between the first two levels, v_t_ = 0 and 1, is only 1.022 cm^−1^, corresponding to 30.64 GHz, which is the value of Δ*E*_01_. Calculating the Boltzmann distribution yields an N_1_/N_0_ ratio of 0.23 to 0.48 for a temperature of 1 to 2 K, i.e., the population of the first excited state v_t_ = 1 is expected to be 23 to 48% with respect to the ground state v_t_ = 0.

### 2.2. Rotational Spectroscopy

#### 2.2.1. Experimental Setups

The rotational spectrum was first observed using the Bologna free jet absorption millimeter-wave spectrometer in the range from 60 to 78 GHz [34,35,36]. Initially, families of intrastate (v_t_ = 0, 1) *μ_c_*-*R*-type transitions such as *J*_10,*x*_ ← (*J*−1)_9,*x*_ and *J*_9,*y*_ ← (*J*−1)_8,*y*_ (with *J* up to 18) were measured and fitted, followed by many intrastate *μ_a_*-*R*-type transitions and several (unexpected, because only *μ_b_* is inverting) interstate *μ_a_*-*R*-type transitions (see the explanation in a following section) [37]. The estimated uncertainty for the measurements is about 50 kHz. The sample was heated to 65–70 °C while a stream of argon with a backing pressure of *P*_a_ = 18 kPa flowed over it. The mixture was then expanded to about *P*_b_ = 0.5 Pa through a 0.3 mm diameter pinhole nozzle. The estimated rotational temperature of the molecules in the jet is 5−10 K. Later on, extensive measurements were made with two molecular jet Fourier transform microwave (MJ-FTMW) spectrometers, one in Bologna [38] (the assignment of the E0 state was made in a way similar to that described in the next section for Aachen) and one in Aachen [39]. Using the scan mode (50 co-added free induction decays per each step in a 250 kHz step width), a broadband scan was recorded from 8 to 14 GHz with the Aachen MJ-FTMW spectrometer. The substance, with a stated purity of 99%, was purchased from Sigma Aldrich, Taufkirchen, Germany, and was used without further purification. Under helium-stream, the phenyl acetate—He mixture entered the vacuum chamber under a backing pressure of 200 Pa. All signals appeared as doublets due to the Doppler effect and were re-measured with higher resolution and under an increased number of free induction decays. The measurement accuracy of both MJ-FTMW spectrometers is about 2 kHz [40]. Eventually, line broadening occurred as a result of unresolved splittings arising from proton hyperfine structures or spin-couplings.

#### 2.2.2. Assignment of the v_t_ = 0, A Species (A0)

Using a rigid rotor model with the values for the rotational constants calculated at the MP2/6-311++G(d,p) level, the *R*-branch *J*+1_1*J*_ ← *J*_0*J*_ and the *Q*-branch *J*_2,*J−2*_ ← *J*_1*J*_ with *J* ≤ 5 were assigned from the survey scan recorded with the Aachen MJ-FTMW spectrometer, independently from the assignment in the millimeter-wave range. The first fit attempts using only three rotational constants yielded a root-mean-squares (rms) deviation of 1.7 MHz. After more lines with higher *J* and *K_a_* from the scan were included in a fit containing 20 lines, five centrifugal distortion constants could be floated, and the rms deviation decreased to 1.24 MHz, which is still extremely large compared to the experimental accuracy of 2 kHz of the MJ-FTMW spectrometer. However, when simulating the spectrum with the fitted rotational and centrifugal distortion constants and comparing it to the measured spectrum, as shown in Figure 7, we were confident that the assignment was correct. Using predictions with the fitted parameters, we found some more lines outside the 8–14 GHz scan area and achieved a fit with 36 lines and an rms deviation of 1.3 MHz.

This struggle in fitting a species using a rigid rotor model supplemented by centrifugal distortion corrections is extremely similar to the fitting procedure in pinacolone [41] and phenyl formate [25]. As the double minimum potential suggests a strong Coriolis coupling in phenyl acetate, a new fit was carried out using the *spfit* program, which can implement Coriolis cross terms. Adding *E* (corresponding to Δ*E*_01_), *E_J_* (multiplying ***J***^2^), *F_bc_* (multiplying ***J****_b_**J**_c_* + ***J****_c_**J**_b_*), and *F_ab_* (multiplying ***J****_a_**J**_b_* + ***J****_b_**J**_a_*) in the fit, the rms decreased to 1.7 kHz, and a total of 80 v_t_ = 0, A species (A0) lines could be assigned in the microwave range. In addition, the assignments of the microwave lines were secured with extensive interlocking combination difference loops, as illustrated in Figure 8. The fit is shown as Fit A0 in Appendix A with a list of all fitted frequencies along with their residuals.

#### 2.2.3. Assignment of the v_t_ = 1, A Species (A1)

After the A0 species was assigned, we found a series of satellite lines close to the strongest A0 species lines. From our experiences with the microwave spectrum of phenyl formate [25], we suspected these transitions to be those of the v_t_ = 1, A species (A1). In a second step, 20 A1 lines could be identified and included in a fit where the rotational and centrifugal distortion constants were fitted separately for the v_t_ = 0 and 1 states (Fit A0/A1 MW in Appendix A). The rms deviation only increased slightly to 3.6 kHz. The effective Hamiltonian:(2)$$H=\sum_{v=0}^{1}v\rangle \left(H_{r}^{v}+H_{∆}^{v}\right)$$
was applied where 0⟩ and 1⟩ represent the symmetric and the anti-symmetric torsional states, respectively, unifying the following operators that describe:
(i)the overall rotation including quartic centrifugal distortion constants:(3)$$H_{r}^{v}=A_{v}J_{a}^{2}+B_{v}J_{b}^{2}+C_{v}J_{c}^{2}-D_{J,v}J^{4}-D_{JK,1}J^{2}J_{a}^{2}-D_{K,1}J_{a}^{4}+d_{1,1}J^{2}\left(J_{+}^{2}\mp J_{-}^{2}\right)+d_{2,0}\left(J_{+}^{4}+J_{-}^{4}\right)$$(ii)the torsional splitting between the 0⟩ and the 1⟩ energy levels:(4)$$H_{∆}^{v}=vE$$(iii)and the Coriolis interaction:(5)$$H_{c}=F_{ab}\left(J_{a}J_{b}+J_{b}J_{a}\right)+F_{abJ}J^{2}\left(J_{a}J_{b}+J_{b}J_{a}\right)+F_{bc}\left(J_{b}J_{c}+J_{c}J_{b}\right)+F_{bcJ}J^{2}\left(J_{b}J_{c}+J_{c}J_{b}\right)$$

The A0/A1 measurements in the microwave range with significantly fewer v_t_ = 1 lines compared to the number of v_t_ = 0 lines were then combined with those in the millimeter-wave range from 60 to 78 GHz, where 40 lines were observed, almost equally distributed between the v_t_ = 0 and 1 states. We could also assign 26 *a*-type interstate transitions v_t_ = 0 ← 1 or 1 ← 0 (note that some of them are blended). The fit is shown as Fit A0/A1 MW+mmw in Appendix A.

#### 2.2.4. Assignment of the v_t_ = 0, E Species (E0)

Assigning the E0 state was a tedious task because the *spfit*/*spcat* program suite was not useful at the initial stage for assigning the internal rotation E species lines and the *XIAM* program could not deal with the tunneling motion of the phenyl ring. We first used *XIAM* to predict a theoretical microwave spectrum for the scan region (8–14 GHz) with the rotational constants of Fit A0 given in Appendix A, the *V*_3_ potential term, and the angles between the internal rotor axis and the principal axes of inertia obtained from calculations at the MP2/6-311++G(d,p) level. The most intense *b*- and *c*-type $J_{K_{a}^{\prime }K_{c}^{\prime }}^{\prime }$ ← $J_{K_{a}K_{c}}$ = 4_13_ ← 3_03_, 5_14_ ← 4_04_, 6_16_ ← 5_05_, 2_20_ ← 1_10_, 3_21_ ← 2_11_, 2_21_ ← 1_11_, 3_22_ ← 2_12_ lines could be identified, being of similar intensity as the corresponding A0 species. Starting from these assignments, we explicitly searched for the same combination difference loops as those found for the A0 species (see Figure 8). After a sufficient number of lines had been assigned, a separate fit of the E0 species was performed with *spfit* using the same parameters accounting for the ring tunneling motion (*E*, *E_J_*, *F_ab_*, and *F_bc_*). In addition, two odd power order parameters, *D_a_* and *D_c_*, as well as their higher-order parameters, *D_aK_*, *D_cK_*, *D_cJK_*, and *D_cKK_* [42], are required to fit the E0 state separately, as implemented in the $H_{op}$ operator, as follows:(6)$$H_{op}=\left(D_{a}+D_{aK}J_{a}^{2}\right)J_{a}+\left(D_{c}+D_{cK}J_{c}^{2}+D_{cKK}J_{c}^{4}\right)J_{c}$$

We achieved an rms deviation of 10.6 kHz for 69 E0 lines. We were confident about the assignments, but due to the low number of lines, no attempts were made to further reduce this deviation by adding more parameters. The fit is shown as Fit E0 in Appendix A with a list of all fitted frequencies along with their residuals. To access the methyl torsional barrier, the A0 and E0 species lines were input into *XIAM*. Similar to the case of the A0 species, we obtained an extremely high rms deviation of 6.9 MHz. The *V*_3_ potential was determined to be 136 cm^−1^. The *XIAM* fit is given in Appendix A along with the observed-minus-calculated values.

#### 2.2.5. Assignment of the v_t_ = 1, E Species (E1)

The predictive power of Fit A0/A1 and Fit E0 enabled us to exclude all lines belonging to the A0, A1, and E0 species from the recorded scan. It was then possible to identify some E1 species lines corresponding to the most intense E0 lines, which are of similar intensity as those of the A1 species, as shown exemplarily for the 2_21_ ← 1_11_ transitions in Figure 9. Eventually, we found 17 E1 lines and fitted them together with the E0 lines using *spfit*. This fit is given as Fit E0/E1 in Appendix A. Similar to Fit A0/A1, the rotational and centrifugal distortion constants were fitted separately for the E0 and E1 species. Odd power order parameters are required, as in the case of Fit E0. They are fitted separately for the E0 and E1 species. Due to the low intensity of the E1 lines, we could not close any combination difference loops. Consequently, though we are confident about most of the E1 assignments, conclusive proof is lacking. The fit chosen for Appendix A has an rms deviation of 358.8 kHz, but we had to take out 9 E0 lines with *K_a_* = 3. Therefore, this high rms deviation may arise from the limited number of fit parameters. The frequency list is also given in Appendix A.

#### 2.2.6. Global Fitting of the A0, E0, A1, E1 Sub-States

We then fit the rotational transition frequencies of all four states, A0, A1, E0, and E1, simultaneously, considering, in principle, all the interactions between the four sublevels of the vibrational ground state. A0 will interact with A1 and E0 via Coriolis and *V*_3_ internal rotation couplings, respectively. Similarly, the other three states interact with two other states. The results of this global fit are reported in Table 1. In the fit, the value of Δ*E*_A0/E0_ was fixed at 16 GHz, corresponding to the *V*_3_ barrier determined by *XIAM*.

Except for *D_K_*, the values of the centrifugal distortion constants are not so homogeneous among the four states. This can be due to the fact that not all the interactions are taken into account. For example, some “local” interactions of rotational states belong to different vibrational sublevels, and their effects can be “included” in the so-called “pseudo-centrifugal distortion constants”, allowing accurate fittings to the experimental uncertainty of the measured frequencies. The output of the *spfit* program leading to the parameters of Table 1 is reported in Appendix A. Similar to the case of Fit E0/E1 in Appendix A, we could not include the 9 E0 lines with *K_a_* = 3 and also had to take out 5 E1 lines. We note that a number of v_t_ = 0 ⟷ 1 interstate *μ_a_*-R-type transitions have been used in the fit, while only *μ_b_*-type interstate transitions should be observed. These “forbidden” transitions originate from an accidental but systematic *μ_b_*-mixing of the v_t_ = 0, *J_7,x_* and the v_t_ = 1, *J_6,y_*, which have very similar energies for each *J*. As an example, for *J* = 14, the energies of the degenerate levels v_t_ = 0, 14_7,7_, and 14_7,6_ are about 303 GHz, being the same as the value of the v_t_ = 1, 14_6,8_, and 14_6,7_ degenerate levels. Their mixing will give a partial *μ_b_*-character to the reported, apparently pure *μ_a_*-type transitions. Moreover, for the E0 and E1 species, about half of the lines are intrastate *μ_b_*-transitions, while due to the kind of motion, only interstate *μ_b_*-transitions are allowed. For the E species, the *K_a_* and *K_c_* quantum numbers have no meaning for those perturbation allowed transitions, because they only indicate the order of energy.

From the value of the parameters *D_a_* or *D_c_* of the E0 state, it is possible to estimate the *V*_3_ barrier to internal rotation of the methyl group when *ac* represents a molecular plane of symmetry [43,44,45]. This is not the case of phenyl acetate, but due to the low barrier to inversion at *α* = 90°, the vibrational wavefunctions of v_t_ = 0 are exactly symmetric with respect to the *ac* plane. In addition, when trying to fit the *D_b_* parameter, it was immediately set to zero. For this reason, we believe that the method to estimate *V*_3_ from *D_a_* (or *D_c_*) is applicable for phenyl acetate [29,43].

We first set a system of the following two equations:(7)$$D_{a}=\frac{A}{r}W_{\nu \sigma }^{\left(1\right)}\lambda_{a}\;\mathrm{and}$$
(8)$$D_{c}=\frac{C}{r}W_{\nu \sigma }^{\left(1\right)}\lambda_{c}$$
to calculate the first-order perturbation coefficients
$W_{0,E}^{\left(1\right)}$ [46] through the relations, as follows:(9)$$W_{0,E}^{\left(1\right)}=\frac{D_{g}}{F\rho_{g}}\;\;\;\;\;\;\;\mathrm{with}\;g=a\;\mathrm{or}\;c,$$

$\lambda_{g}=cos\left(∢i,g\right)$, assuming the top inertial defect *I_α_* = 3.2 uÅ*^2^*. Since the *ac* plane can be considered as a symmetry plane (∢i,a) + (∢i,c) = 90°; therefore, *λ_a_* and *λ_c_* are dependent, leading to only two unknowns. The solution of the two-equation system provides (∢i,a) = 22.1°, (∢i,c) = 67.9°, and $W_{0,E}^{\left(1\right)}$ = 0.1133.

From the coefficient $W_{0,E}^{\left(1\right)}$, the reduced barrier *s* estimated through Herschbach’s tables is 11.5 [46]. The reduced barrier is related both to the methyl internal rotation barrier
(10)$$V_{3}=\frac{9}{4}F\cdot s$$
and the splitting between the A and E levels, as follows:(11)$${∆}_{0}=\frac{27}{8}Fw_{1}^{v=0}$$
with $w_{1}^{v=0}$ = −0.0275 from Herschbach’s tables. The achieved values are Δ_0_ = 0.5 cm^−1^ and thus *V*_3_ = 143 cm^−1^.

### 2.3. Flexible Model Calculations

The experimental value of the Δ*E*_A0/A1_ vibrational splitting can be used to determine the small barrier at α = 90°. In order to handle a simple expression, we first defined *τ* = α − 90 and used the following double minimum potential:(12)$$V\left(\tau \right)=B_{2}{\left[1-{\left(\tau /\tau_{0}\right)}^{2}\right]}^{2}$$
where *B*_2_ is the barrier at *τ* = 0° and *τ*_0_ is the equilibrium value of the inversion angle. Such an equation is suitable to describe our potential energy function in a small interval that includes the two minima and the “transition state”. The energies and wavefunctions of vibrational states related to the skeletal torsion were calculated numerically with a suitable model [47]. We fixed the parameter *τ*_0_ to the *ab initio* value, calculated *B*_2_, and applied Meyer’s one-dimensional numerical flexible model [47]. The value Δ*E*_A0/A1_ = 36.4 GHz was reproduced when *B*_2_ was set to 68 cm^−1^. In the flexible model calculation, the *τ* coordinate has been considered in the ± 50° range and solved into 41 mesh points [47]. This range is expected to reproduce the low energy part of the potential energy curve of this motion, describing the minima and the barrier at the perpendicular configuration. The value of *B*_2_ = 68 cm^−1^ is considerably higher than the value obtained by the Hamiltonian in Equation (1). This discrepancy can be due to the fact that, with the numerical method, the value of the reduced constant of the motion *F* is calculated for every point, and it undergoes significant variations in the considered range.

## 3. Discussion

The A and E species lines belonging to the v_t_ = 0 tunneling species of phenyl acetate were assigned. The initial assignments of both species are secured by an extensive interlocking network of combination difference loops. These A0 and E0 states could be fitted separately using the *spfit* program to rms deviations close to the measurement accuracy if parameters accounting for the ring tunneling motion are considered (see Appendix A). This “local” approach has been proven to be very powerful as assignment guidance in many previous studies where methyl internal rotations with low torsional barriers are present [42,48,49,50,51].

Though it is possible to derive the value of the barrier to internal rotation from the fitted odd power parameters *D_a_* and *D_c_*, the obtained barrier (*V*_3_ = 143 cm^−1^) might not be so accurate due to several approximations [29]. The *V*_3_ barrier was also deduced by a *XIAM* fit, including the A0/E0 species. Since *XIAM* does not take into account the interactions of A0 and E0 with A1 and E1, the standard deviation of the fit is terribly high. However, the value of *V*_3_ = 136 cm^−1^ obtained from this fit is extremely close to the value of 135.3498(38) cm^−1^ found for isopropenyl acetate, where the isopropenyl group is also titled out of the O-(C=O)-C plane by an angle of about 70° [14]. As summarized in a review by Nguyen and Kleiner [52], a torsional barrier of the acetyl methyl group CH_3_-COO is almost invariant at 100 cm^−1^ for *α*,*β*-saturated acetates [5,6,7,8,9,10]. In *α*,*β*-unsaturated acetates with the two current representatives vinyl acetate [11,12] and two isomers of butadienyl acetate [13], the conjugation over the double bond(s) increases the barrier height to 150 cm^−1^. In both isopropenyl acetate and phenyl acetate, the double bond is not located in the O-(C=O)-C plane, leading to a less effective conjugation. The torsional barrier of about 136 cm^−1^, between 100 cm^−1^ and 150 cm^−1^, reflects well this observation.

The v_t_ = 1 spectrum is significantly less intense than the v_t_ = 0 one (see Figure 9, for example, note that the intensity is on a logarithmic scale to better visualize the weaker A1 and E1 species lines). The number of line ratio is N_1_/N_0_ ≈ 1/4. The tunneling Δ*E* parameter could be determined and has a value of 36.4 GHz for the A0/A1 and 33.5 for the E0/E1 fit (see Table 1). Both values are close to the value of 30.6 GHz obtained from calculations using a simple two-top torsional Hamiltonian of the CH_3_-COO group and the phenyl ring. Compared to phenyl formate [25], where the ring tunneling is respected to an H-COO frame instead of the heavier CH_3_-COO frame as in the case of phenyl acetate, the Coriolis splitting Δ*E*_01_ of 46.2231(25) GHz of the former is significantly larger. On the other hand, the Δ*E*_A0/A1_ value of phenyl acetate is an order of magnitude larger than the value of 3.732027(43) GHz in *E*-phenylformamide [24]. The smaller tilt angle of 40° of the phenyl ring out of the (NH)(CO) plane in *E*-phenylformamide is mainly responsible for this Coriolis splitting being drastically lower than in phenyl formate or phenyl acetate.

The coupled LAMs observed in the microwave spectrum of phenyl acetate are very similar to those found in pinacolone [41], where in the latter molecule, a methyl internal rotation is coupled with the tunneling motion of the *tert*-butyl group. We show here that a model, often successful, as implemented in *XIAM,* fails in some specific cases. Even a very powerful and flexible program such as *spfit/spcat* encounters problems when tunneling motions between a double minimum potential come into play with a low-barrier methyl torsion. Kleiner’s BELGI-hybrid program may be suitable to treat this problem in another global approach, taking into account the interactions of all four sub-states [53]. After the A0, A1, E0, and E1 species were assigned, all lines remaining in the broadband scan are of weak intensity and are neglected.

## Figures and Tables

**Figure 1 molecules-27-02730-f001:**
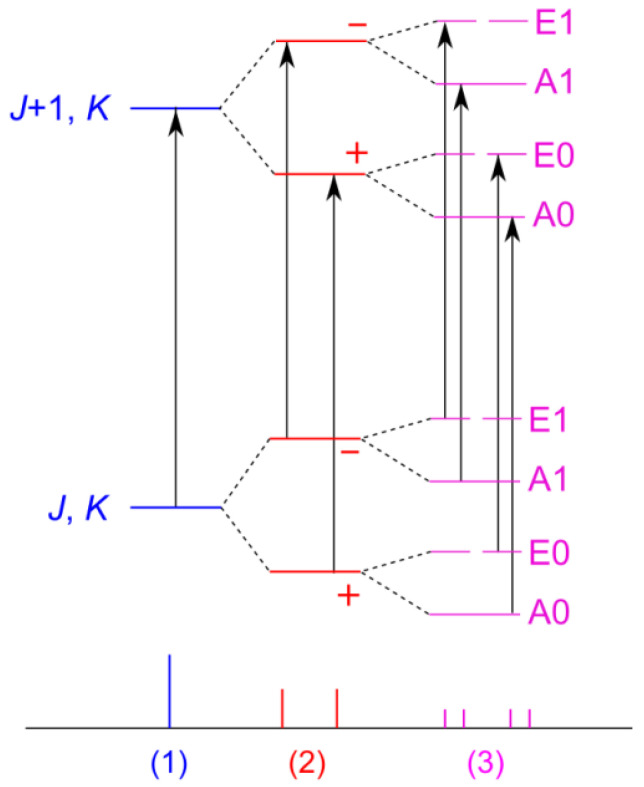
Non-scaled energy levels and microwave signal splittings for different types of molecules. (1) Asymmetric rigid rotor (e.g., phenetole [16]), (2) One two-fold tunneling motion (e.g., phenyl formate with the tunneling of the phenyl ring [25]), (3) One two-fold tunneling motion and a methyl internal rotation (e.g., phenyl acetate with the tunneling of the phenyl ring and the internal rotation of the acetyl methyl group, this work). Rotational *a*-type transitions are indicated as black arrows. Please note that, for sake of simplicity, the symmetric top *J*, *K* label is used to indicate the rotational levels, rather than the correct *J*, *K_a_*, *K_c_* one.

**Figure 2 molecules-27-02730-f002:**
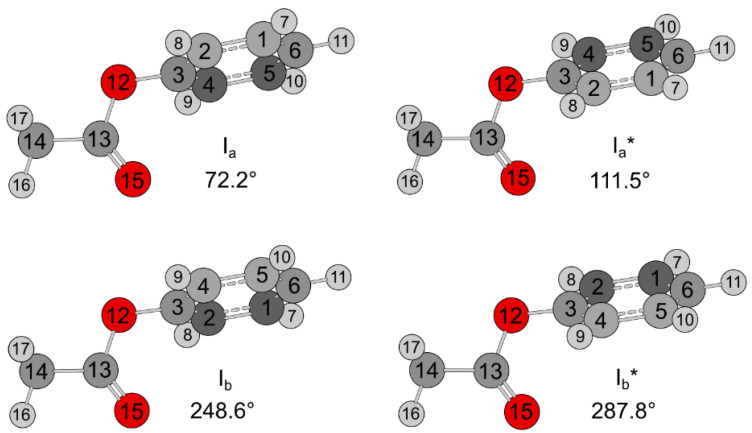
Four minima equivalent in energy at *E* = −458.9579885 E_h_ of the most stable conformer I (*entgegen*) of phenyl acetate with C_1_-symmetry. The geometries were fully optimized at the MP2/6-311++G(d,p) level of theory. The dihedral angle *β* = ∠(C_3_-O_12_-C_13_-C_14_) describes the *entgegen* orientation of the acetyl group and is 180°, while *α* = ∠(C_13_-O_12_-C_3_-C_2_) corresponds to the orientation of the phenyl ring, which is 72.2° for I_a_, 111.5° for I_a_^*^, 248.6° for I_b_, and 287.8° for I_b_^*^. Note that *α* ≈ 90° or 270° at the transition states, corresponding to a tunneling angle of about 40° from one minimum to the other.

**Figure 3 molecules-27-02730-f003:**
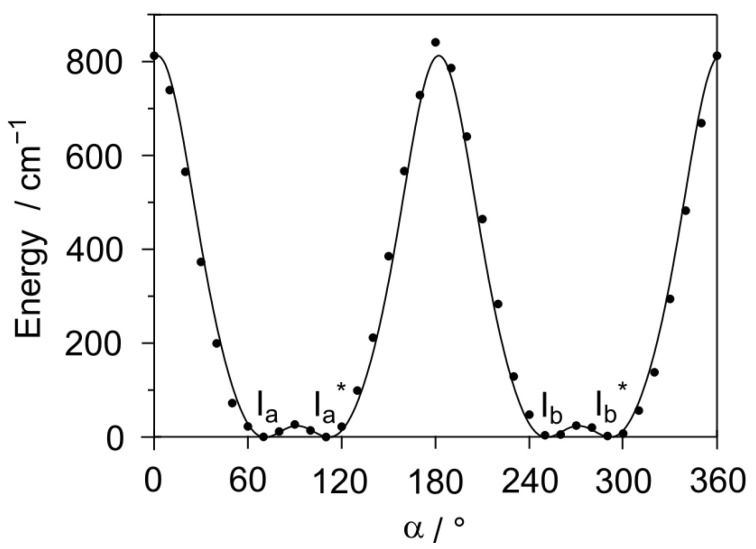
Potential energy curve of phenyl acetate obtained by varying the dihedral angle *α* = ∠(C_13_-O_12_-C_3_-C_2_) in 10° steps. The two double minima correspond to the I_a_/I_a_* and I_b_/I_b_* pairs. The barrier to convert I_a_ in I_a_^*^ or I_b_ in I_b_^*^ is 20.65 cm^−1^; that to convert I_a_ in I_b_ or I_a_^*^ in I_b_^*^ is 837.45 cm^−1^. The minima are at *α* = 70°, 110°, 250°, and 290° with an energy of *E* = −458.9579885 E_h_. All other energy values are relative to this value.

**Figure 4 molecules-27-02730-f004:**
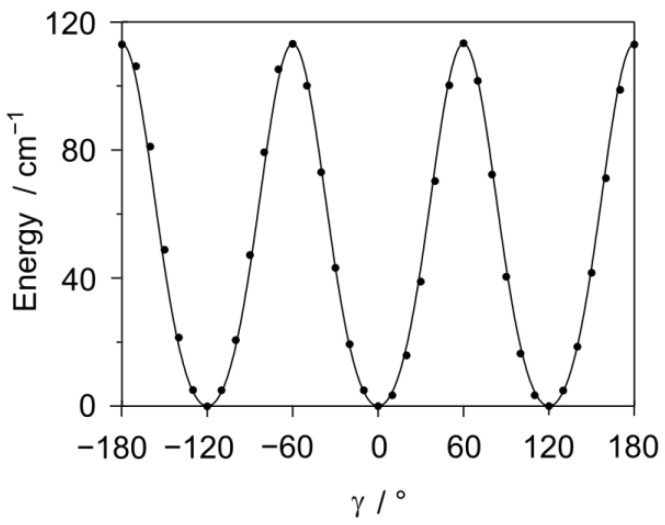
The threefold potential obtained by rotating the methyl group about the C_13_-C_14_ axis (variation of the dihedral angle *γ* = ∠(O_15_-C_13_-C_14_-H_16_)). The optimized structure of conformer I_a_ served as the input structure. Three equivalent minima were obtained for *γ* = −120°, 0°, and 120°. The torsional barrier of about 113 cm^−1^ is relatively low, suggesting large torsional splittings in the microwave spectrum. Energies are given relative to *E*_min_ = −458.9577608 E_h_.

**Figure 5 molecules-27-02730-f005:**
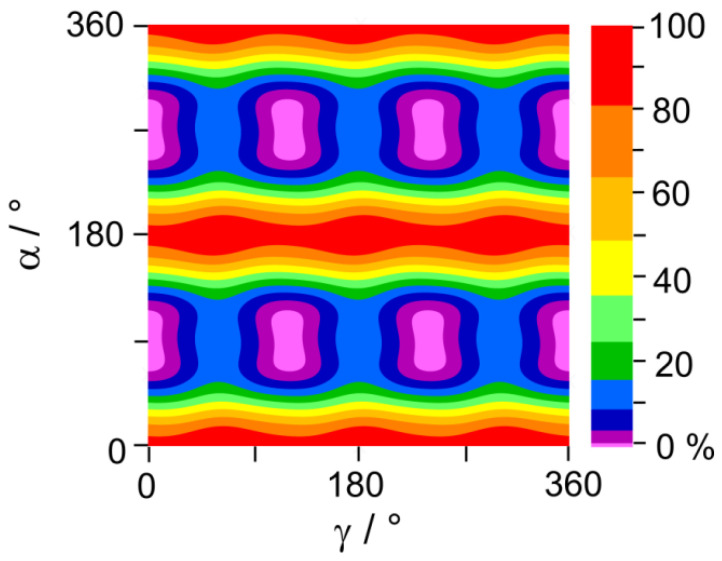
The potential energy surface of phenyl acetate calculated at the MP2/6-311++G(d,p) level of theory in dependence of the dihedral angles *α* and *γ*. The energies are given in percentage color code, relative to *E*_min_ = −458.9579660 E_h_ (0%), and *E*_max_ = −458.9533836 E_h_ (100%). The relative energy of the maximum is about 12 kJ/mol. Note that there are more colors in the lower 50% region.

**Figure 6 molecules-27-02730-f006:**
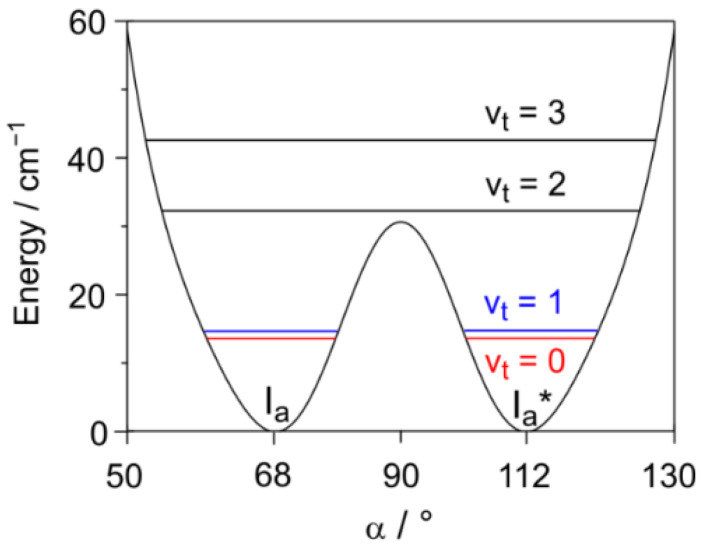
An enlarged portion from 50° to 130° of the potential energy curve given in Figure 3. The doubly degenerate lowest tunneling energy levels v_t_ = 0, 1, 2, 3 are indicated by horizontal lines.

**Figure 7 molecules-27-02730-f007:**
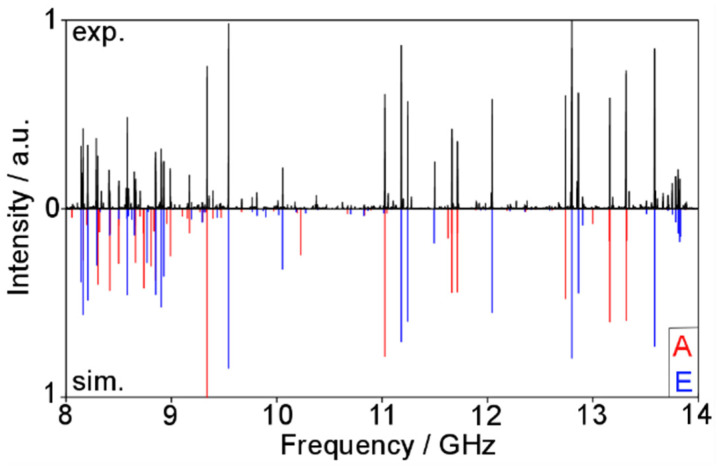
Upper trace: The recorded survey spectrum of phenyl acetate reaching from 8 to 14 GHz. Lower trace: The theoretical spectrum reproduced with the parameters obtained from Fit A0/E0 (see text). The A0 species transitions are marked in red, the E0 species in blue.

**Figure 8 molecules-27-02730-f008:**
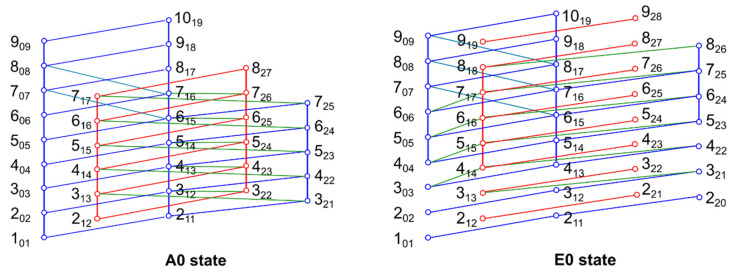
Schematic illustration of some rotational energy levels $J_{K_{a}K_{c}}$ (open circles) of the A0 and E0 species of phenyl acetate measured with the Aachen FTMW spectrometer. Solid lines connecting the circles indicate transitions that are checked by combination difference loops, which sum to values in good agreement with the expected measurement uncertainty of 2 kHz. These transitions were used for an initial *spfit* fit whose *spcat* predictions have guided the assignments of further lines. Note that for the E0 states, some *a*-type transitions are weak or fall in resonance ranges of the spectrometer and could not be measured. Therefore, some loops are not closed.

**Figure 9 molecules-27-02730-f009:**
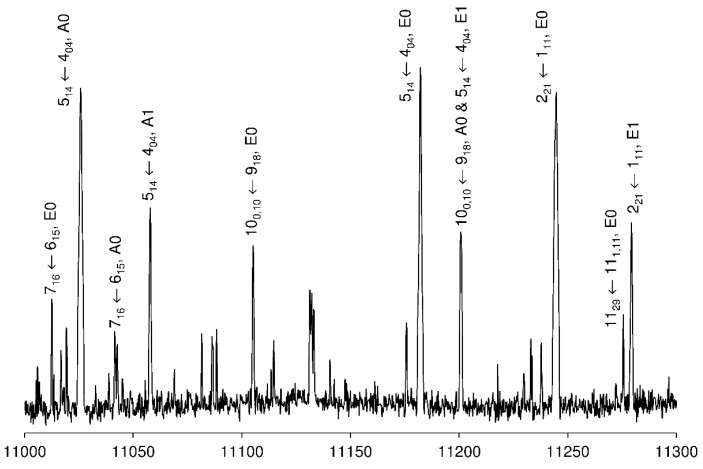
A portion of the scan of phenyl acetate in the frequency range from 11,000 to 11,300 MHz recorded by overlapping spectra with 50 co-added decays per each spectrum. The logarithmic intensities are in arbitrary units. The assigned lines are labeled with the corresponding rotational quantum numbers and tunneling species.

**Table 1 molecules-27-02730-t001:** Molecular parameters obtained from a simultaneous fitting of the A0, A1, E0 and E1 sub-state lines of phenyl acetate using the program *spfit*.

Par. ^a^	Unit	A0	A1	E0	E1
*A*	MHz	3637.78667(33)	3640.89405(39)	3622.9532(38)	3627.018(13)
*B*	MHz	803.88947(10)	803.39084(22)	803.80593(60)	803.3164(40)
*C*	MHz	750.97790(11)	749.94839(46)	750.94959(64)	749.8638(26)
*D_J_*	kHz	0.2243(17)	−0.0429(32)	−1.582(53)	2.99(18)
*D_JK_*	kHz	2.3359(92)	2.451(13)	7.78(77)	−35.56(19)
*D_K_*	kHz	0.193(13)			
*d* _1_	kHz	−0.00130(24)	0.0366(18)	0.1360(13)	--
*d* _2_	kHz	0.1583(13)	−0.0142(23)	1.509(48)	--
Δ*E*	GHz	36.40881(32)		33.533(75)	
*F_bc_*	MHz	28.0660(25)		25.07(11)	
*F_bcK_*	kHz	0.1877(69)		−0.334(53)	
*F_ab_*	MHz	78.5913(19)		76.77(12)	
*F_abJ_*	kHz	3.307(47)		0.0139(10)	
*F_abK_*	kHz	1.009(14)		0.01385(41)	
*D_a_*	MHz			401.8794(20)	380.792(12)
*D_c_*	MHz			32.6882(22)	36.129(92)
*D_aJ_*	MHz			−0.14776(32)	−0.0762(12)
*D_cJ_*	MHz			2.376(76)	−8.860(98)
*D_aK_*	kHz			7.596(90)	−11.42(27)
*D_cK_*	kHz			−7.73(27)	−9.49(97)
*N* ^b^		240			
σ/σ_exp_ ^c^		1.22			

^a^ All parameters refer to the principal axis system. Watson’s S reduction in I^r^ representation was used. ^b^ Numbers of lines included in the fit. ^c^ Reduced deviation of the fit when setting the millimeter-wave and microwave measurement errors to 80 kHz and 3 kHz, respectively.

## Data Availability

Data are contained within the article and Appendix A.

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
