# Peer review of "Skeletal Torsion Tunneling and Methyl Internal Rotation: The Coupled Large Amplitude Motions in Phenyl Acetate"

_molecules, 2022, doi:10.3390/molecules27092730_

Round 1

Reviewer 1 Report

This manuscript presents a combined computational and spectroscopic study of methyl internal rotation in phenyl acetate. The work seems competently done, the methods are adequate, and the conclusions seem supported by the results.

The interest to a broad audience is probably feble, as it addresses a localized question and a very specific problem. However, this specialized works are on the foundations of spectroscopy and the combined use of spectroscopic and computational techniques is meritorious.  And, as mentioned above, the work seems competently done, with a detailed description of the methods used and limitations of each approach. For these reasons, it deserves publication on “Molecules”.

It should be mentioned, however, that this Reviewer has expertise on computational methods and free jet spectroscopy techniques, but not on fine details of rotational spectroscopy analysis.

Reviewer 2 Report

In this manuscript, the authors studied the rotational spectrum of phenyl acetate by using a free jet 13 absorption millimeterwave spectrometer in the range from 60 to 78 GHz and two pulsed jet Fourier 14 transform microwave spectrometers covering a total frequency range from 2 to 26.5 GHz. The 15 features of two large amplitude motions were studied. It provides useful information for industrial applications and is an organized and well-written manuscript and I recommend its publication in Molecules after minor revisions:

  1. The dimensions of graphics should be properly compressed.
  2. The conclusion should be properly compressed. In addition, the conclusion should not be a list of experimental results, but the scientific laws.
  3. The reference format is chaotic. The reference style should be given in the format requested by the
  4. English writing needs significant improvement. There are some grammatical errors and spelling mistakes in this manuscript.

Reviewer 3 Report

In the manuscript, the authors present their extensive study of the structure and the couplings present in phenyl acetate using microwave spectroscopy. The paper is interesting, well-written, and nicely shows how the authors arrived at their conclusions. Thus, it can be accepted after some minor corrections. In the following, I add a few suggestions and spotted spelling mistakes:

In the manuscript, it is mentioned that in most cases “medium-sized molecules containing a phenyl ring have […] a planar heavy-atom structure as the most stable one”. In this discussion, one might add that this is only the case as long as no through-space effects occur as in the case of 2-phenylethylamine [Godfrey et al. J. Am. Chem. Soc. 117, 8204 (1995)]. In this system, there is not a single structure where all heavy atoms are within the aromatic plane. This system has also been studied with rotational resolution by the group of Alonso [Lopez et al., Phys. Chem. Chem. Phys. 9, 4521 (2007)].

If I understand the section about the assignment of the rotational lines correctly, the goodness of the fit increased considerably when using the SPFIT/SPCAT program. If this is the case, why does Fig. 7 show the results from the inferior fit? This should be adapted to illustrate the results of the best fit there is.

Line 65: This is the first place where the term “Coriolis splitting” occurs in the manuscript. Hence, it is not clear to what the sentence “… from the above-mentioned Coriolis splittings” refers.

The scheme of the rotational levels in Fig. 1 is remarkably clear and intuitive.

Line 50: “how high” instead of “how height”

Line 190: Please add brackets around the argument of the cosine to clarify this expression.

Line 290: It would be helpful if the authors could add a brief explanation in the caption about which quantum numbers are shown in Fig. 8. I guess these are Ka and Kc, but it is never mentioned explicitly. The same applies to the transitions listed in line 314.
